# Thyroid Hormone Replacement Therapy Is Associated with Longer Overall Survival in Patients with Resectable Gastroesophageal Cancer: A Retrospective Single-Center Analysis

**DOI:** 10.3390/cancers13205050

**Published:** 2021-10-09

**Authors:** Hannah C. Puhr, Thorsten J. Reiter, Mohamed El-Mahrouk, Lena Saliternig, Peter Wolf, Maximilian J. Mair, Ariane Steindl, Matthias Paireder, Reza Asari, Sebastian F. Schoppmann, Anna S. Berghoff, Matthias Preusser, Aysegül Ilhan-Mutlu

**Affiliations:** 1Department of Medicine I, Division of Oncology, Medical University of Vienna, 1090 Vienna, Austria; hannah.puhr@meduniwien.ac.at (H.C.P.); n01629138@students.meduniwien.ac.at (T.J.R.); n01618510@students.meduniwien.ac.at (M.E.-M.); n01613731@students.meduniwien.ac.at (L.S.); maximilian.mair@meduniwien.ac.at (M.J.M.); ariane.steindl@meduniwien.ac.at (A.S.); anna.berghoff@meduniwien.ac.at (A.S.B.); matthias.preusser@meduniwien.ac.at (M.P.); 2Comprehensive Cancer Center Vienna, Medical University of Vienna, 1090 Vienna, Austria; matthias.paireder@meduniwien.ac.at (M.P.); reza.asari@meduniwien.ac.at (R.A.); sebastian.schoppmann@meduniwien.ac.at (S.F.S.); 3Department of Medicine III, Division of Endocrinology and Metabolism, Medical University of Vienna, 1090 Vienna, Austria; peter.wolf@meduniwien.ac.at; 4Department of Surgery, Medical University of Vienna, 1090 Vienna, Austria

**Keywords:** gastrointestinal neoplasms, stomach neoplasms, esophageal neoplasms, thyroid gland, thyroid diseases

## Abstract

**Simple Summary:**

Thyroid hormones are surmised to be associated with cancer. However, so far, there is no sufficient data on the association of thyroid hormone status as well as replacement therapy in patients with upper gastrointestinal cancer Thus, we analyzed clinical as well as endocrinological parameters of 865 patients with resectable gastroesophageal cancer treated at the Medical University of Vienna, which is a large representative European cohort. A tendency towards prolonged overall survival in hypothyroid patients (euthyroid, *n* = 647: median OS 29.7 months; hyperthyroid, *n* = 50: 23.1 months; hypothyroid, *n* = 70: 47.9 months; *p* = 0.069) as well as a significant positive correlation of thyroid hormone replacement therapy with the overall survival was observed (without, *n* = 53: median OS 30.6 months; with, *n* = 67: 51.3 months; *p* = 0.017). Thus, thyroid disorders and their therapeutic interventions might pose as potential prognostic tools and further prospective analyses are warranted.

**Abstract:**

Introduction: As thyroid hormones modulate proliferative pathways it is surmised that they can be associated with cancer development. Since the potential association of gastroesophageal cancer and thyroid disorders has not been addressed so far, the aim of this study was to investigate the association of thyroid hormone parameters with the outcome of these patients, so novel prognostic and even potentially therapeutic markers can be defined. Material and Methods: Clinical and endocrinological parameters of patients with resectable gastroesophageal cancer treated between 1990 and 2018 at the Vienna General Hospital, Austria, including history of endocrinological disorders and laboratory analyses of thyroid hormones at first cancer diagnosis were investigated and correlated with the overall survival (OS). Results: In a total of 865 patients, a tendency towards prolonged OS in hypothyroid patients (euthyroid, *n* = 647: median OS 29.7 months; hyperthyroid, *n* = 50: 23.1 months; hypothyroid, *n* = 70: 47.9 months; *p* = 0.069) as well as a significant positive correlation of thyroid hormone replacement therapy with the OS was observed (without, *n* = 53: median OS 30.6 months; with, *n* = 67: 51.3 months; *p* = 0.017). Furthermore, triiodothyronine (T3) levels were also associated with the OS (median OS within the limit of normal: 23.4, above: 32.4, below: 9.6 months; *p* = 0.045). Conclusions: Thyroid disorders and their therapeutic interventions might be associated with the OS in patients with resectable gastroesophageal cancer. As data on the correlation of these parameters is scarce, this study proposes an important impulse for further analyses concerning the association of thyroid hormones with the outcome in patients with gastroesophageal tumors.

## 1. Introduction

Cancers of the upper gastrointestinal (GI) tract including gastric, esophageal, and gastroesophageal junction (GEJ) cancer are among the most common and deadly malignant diseases worldwide [1]. Survival in these patients is generally poor and reliable prognostic markers are scarce, especially in European cohorts [2]. Given that clinical outcomes have been found to vary substantially even within the same stage groups, new prognostic tools are needed. As thyroid hormones modulate proliferative and angiogenic pathways, it is surmised that they may be associated with cancer development and progression [3,4,5]. Furthermore, the loss of normal functions of thyroid receptors might also play an important role as a contributor to cancer development as well as metastasis [6]. However, data on the association of thyroid hormones and tumors, especially in the GI tract, are scarce and to the best of our knowledge, there is currently no data on the association of thyroid hormones and resectable gastroesophageal cancer. So far, we already investigated the potential survival correlation of thyroid hormones in advanced and metastatic gastroesophageal cancer patients and showed that elevated free thyroxine levels are associated with poorer overall survival (OS) in this patient cohort [7]. Thus, the aim of this study was to investigate the association of thyroid hormone parameters with the outcome of patients with resectable gastroesophageal tumors, so potential, novel prognostic markers can be defined.

## 2. Materials and Methods

### 2.1. Study Design

This analysis was a single-center retrospective study assessing the prognostic association of the thyroid hormone status at the time of first diagnosis as well as patients’ history of thyroid disorders and hormone replacement therapy with the course of disease of patients with resectable upper GI cancers.

### 2.2. Patients Recruitment

All patients who had a histologically proven upper GI cancer and were either diagnosed and/or treated at the Department of Medicine I–Division of Oncology at the General Hospital of Vienna between 1990 and 2018 were included in this analysis. All included patients underwent tumor staging prior to therapy according to the local hospital standard, including history taking, physical examination, routine hematologic tests, upper gastrointestinal endoscopy with histological biopsy, and CT of the chest and abdomen. Patients were treated according to the individual decision of an interdisciplinary tumor board, which ensured the best possible treatment according to the respective standard of knowledge at the time of diagnosis.

Patients were followed up until death according to the hospital or public records or loss to follow-up.

Patients under the age of 18 at the time of diagnosis were excluded as well as patients who were only diagnosed in the General Hospital of Vienna but subsequently treated exclusively in other health institutions.

### 2.3. Data Recruitment

Clinical information including patient demographics, therapy regimens, thyroid hormones, and survival outcome was obtained from hospital chart data. We evaluated endocrinological parameters including thyroid hormones including thyroid-stimulating hormone (TSH), triiodothyronine (T3), thyroxine (T4), free triiodothyronine (fT3), and free thyroxine (fT4). Normal limits were considered to be: TSH 0.44–3.77 μU/mL, T3 0.8–1.8 ng/mL, T4 58–124 ng/mL, fT4 0.76–1.66 ng/dL, and fT3 2.15–4.12 pg/mL. All parameters were determined using standard assays and procedures according to hospital routine.

Thyroid disorders were defined using blood tests including TSH as well as fT4. TSH levels lower than the normal range of 0.44 μU/mL indicate hyperthyroidism and TSH levels higher than the normal range of 3.77 μU/mL indicate hypothyroidism. Whether the dysfunction is clinically relevant is determined by thyroxine levels. If TSH levels are out of the normal range, but thyroxine levels are within the normal range the thyroid disorder is classified as subclinical. If TSH is lower and thyroxine is higher than the normal limit, this indicates clinically relevant hyperthyroidism. If TSH is higher and thyroxine is lower than the normal limit, this indicates clinically relevant hypothyroidism.

### 2.4. Data Safety

Personal data was consecutively numbered and pseudonymized within data collection. Only authorized personnel have access to our database. Clinical data was stored in a password-protected Filemaker© database (FileMaker Inc., Santa Clara, CA, USA) on a protected server of the Medical University of Vienna.

### 2.5. Statistical Analysis

Investigated results were analyzed with the statistical package for the social sciences (SPSS) 20.0 software (SPSS Inc., Chicago, IL, USA). Chi-squared test was utilized for the analysis of the distribution of dichotomized variables. Due to the hypothesis-generating design of the current study no correction for multiple testing was applied [8]. Patients without an event (death) were censored at the date that they were last known to be alive.

Differences between groups were assessed using the Chi square test, the Kruskal–Wallis test, the Mann–Whitney U test, and the Pearson correlation as appropriate.

OS was calculated from the date of the initial diagnosis to the death of the patient or the patient’s last follow-up date. The date of death was evaluated either by hospital chart data or by the records from the Austrian statistical office “Statistics Austria”. Analyses of the OS were conducted with Kaplan–Meier survival estimates with log-rank test and Cox regression. Using log-rank test following parameters were correlated with outcome: gender, body mass index, nicotine, alcohol, family history, localization of cancer, histology, treatment (neoadjuvant versus adjuvant), surgical resection, radiotherapy, known endocrinological disorders. Using Cox regression analysis, the following parameters were correlated with outcome: age, serum TSH, serum ft3, serum ft4, serum T3, serum T4. Furthermore, patients were split into groups with hormone levels below, within, or above the normal range, and their group assignment was correlated with OS using log-rank tests. Two-tailed *p*-values of ≤0.05 were considered to be statistically significant.

## 3. Results

### 3.1. Patient and Tumor Characteristics

Further analysis was performed on 865 patients with resectable gastroesophageal cancer. Demographics and cancer-specific characteristics are shown in Table 1. Twelve patients (1.4%) were in a resectable setting, but the exact stage could not be identified due to missing data (i.e., no endoscopic ultrasound). The distribution of histological subtypes in relation to tumor localization is shown in Appendix A.

A history of a second cancer either before (100 patients, 11.6%) or at the same time (64 patients, 7.4%) of the diagnosis of gastroesophageal cancer was reported in around 19% of the patients. Concerning alcohol consumption, 95 patients were known alcoholics (11.0%), whereas 310 (35.8%) patients did not drink any alcohol and 311 (36.0%) patients had a moderate consumption. Smoking was recorded in 414 (47.9%) patients. Concerning treatment options, 455 (52.6%) patients received chemotherapy either in an adjuvant, neoadjuvant, or later on in a palliative setting. In total, 775 (89.6%) patients received a surgical tumor removal, whereas radiation therapy was administered in 183 (21.1%) patients.

### 3.2. Overall Survival in Regard to Patient and Tumor Characteristics

At the time of data cut-off (December 2019), 684 (79.1%) patients were dead, 181 (20.9%) patients were either alive or lost to follow-up. The median overall survival (OS) of the study population was 29.8 months (95% confidential interval (CI) 26.1–33.5).

The correlation of demographics and the baseline characteristics as well as characteristics of the malignancy with survival are shown as *p* values in Table 1. The Kaplan–Meier figures can be seen in Appendix A. Older age is statistically significantly associated with shorter OS (*p* < 0.001; hazard ratio (HR) 1.014). Furthermore, the tumor location (*p* = 0.004), the histology (*p* = 0.001), as well as the tumor stage (*p* < 0.001) have a significant association with the OS.

Since patients were included in this study between 1990 and 2018, we analyzed whether the year of diagnosis had a statistically significant correlation with the OS. The OS increased statistically significantly throughout the years (*p* = 0.025, HR = 1.014, 95% CI 1.002–1.027). However, in a multivariate analysis including stage and the year of diagnosis, the stage remains statistically associated with the OS (*p* < 0.001, HR = 1.625), while the year of diagnosis is not (*p* = 0.485, HR = 1.005).

### 3.3. Thyroid Parameters

The median absolute values of the evaluated thyroid parameters at the time of first diagnosis including TSH, T3, T4, fT3, and fT4 were within the normal range both in the overall and in separate cohorts based on the tumor stage. The results are shown in Table 2. Additionally, boxplots are available in Appendix A to visualize these results in the overall cohort.

At the time of cancer diagnosis, the TSH levels of 209 (24.2%) patients were available. Although median absolute values of TSH were within the normal limit, several patients had TSH levels above or below the normal limit. In 7 patients the TSH levels were above the normal limit, suggesting hypothyroidism, and in 26 patients the levels were below the normal limit, suggesting hyperthyroidism.

In addition, known thyroid disorders in the patients’ history were evaluated; in 50 patients (5.8%) hyperthyroidism and in 70 (8.1%) hypothyroidism was recorded. Six hundred and forty-seven (74.8%) patients did not have a known thyroid disorder at the time of cancer diagnosis. In 98 (11.3%) patients, no data on thyroid disorders was available. Some patients had a history of subclinical thyroid disorders, which is shown in Appendix A.

In regard to tumor stages, the distribution of thyroid disorders is shown in Appendix A.

Concerning therapies for thyroid disorders, 67 (7.7%) patients were taking thyroid hormone substitution therapy at the time of first cancer diagnosis and 41 (4.7%) patients had had a thyroidectomy. Of these 67 patients taking thyroid replacement therapy, 56 (83.6%) had a history of hypothyroidism and 11 (16.4%) had a history of hyperthyroidism (taking thyroid replacement therapy after complete thyroidectomy).

### 3.4. Thyroid Hormones and Their Correlation with the Overall Survival

Concerning thyroid hormone levels, neither TSH nor T3, T4, fT3, or fT4 levels did not significantly correlate with the survival outcome, neither in the overall cohort nor in separate cohorts based on the tumor stage. These results are shown in Table 2.

Furthermore, we analyzed whether the values were within, above, or below the limit of normal and correlated them with the OS. In the overall cohort, T3 (median OS within the normal limit, *n* = 53: 23.4 months, 95% CI 3.9–42.9; above the normal limit, *n* = 1: 32.4 months; below the normal limit, *n* = 6: 9.6 months, 95% CI 7.1–12.1; *p* = 0.045) was statistically significantly associated with the OS. However, TSH (median OS within the normal limit, *n* = 176: 26.0 months, 95% CI 17.0–35.0; above the normal limit, *n* = 7: 47.9 months, 95% CI 24.7–71.1; below the normal limit, *n* = 26: 13.0 months, 95% CI 4.9–21.1; *p* = 0.455), T4 (median OS within the normal limit, *n* = 82: 21.7 months, 95% CI 10.0–33.4; below the normal limit, *n* = 7: 42.5 months, 95% CI 0.7–84.3; *p* = 0.427), fT3 (median OS within the normal limit, *n* = 33: 22.0 months, 95% CI 8.2–35.8; above the normal limit, *n* = 3: 6.3 months, below the normal limit, *n* = 2: 19.7 months; *p* = 0.922) and fT4 (within the normal limit, *n* = 72; above the normal limit, *n* = 7; below the normal limit, *n* = 1; *p* = 0.968; all cases censored) were not. These results are shown in Figure 1.

### 3.5. Thyroid Diseases and Their Correlation with the Overall Survival

The overall survival of patients with a history of thyroid disorders did not differ significantly from euthyroid patients, neither considering the overall cohort (overall cohort: euthyroid, *n* = 647: median OS 29.7 months; 95% CI 26.0–33.4; hyperthyroid, *n* = 50: median OS 23.1 months; 95% CI 10.2–36.0; hypothyroid, *n* = 70: median OS 47.9 months; 95% CI 20.1–75.7; *p* = 0.069) nor separate cohorts based on stages (stage I: euthyroid, *n* = 121: median OS 82.8 months; 95% CI 58.2–107.4; hyperthyroid, *n* = 6: median OS 60.3 months; 95% CI 0–156.6; hypothyroid, *n* = 13: median OS 221.4 months; 95% CI 110.9–331.9; *p* = 0.147; stage II: euthyroid, *n* = 198: median OS 39.4 months; 95% CI 31.7–47.1; hyperthyroid, *n* = 17: median OS 36.5 months; 95% CI 10.7–62.3; hypothyroid, *n* = 21: median OS 65.0 months; 95% CI 37.0–93.1; *p* = 0.429; stage III: euthyroid, *n* = 320: median OS 19.6 months; 95% CI 17.1–22.1; hyperthyroid, *n* = 26: median OS 18.0 months; 95% CI 6.6–29.4; hypothyroid, *n* = 35: median OS 26.7 months; 95% CI 15.8–37.6; *p* = 0.259). However, in both the overall as well as in the separated cohorts, a tendency that hypothyroid patients have a longer OS could be observed. This result is shown in Figure 2.

The use of thyroid replacement therapy was associated with a statistically significant longer overall survival comparing patients with thyroid disorders in the overall cohort (without, *n* = 53: median OS 30.6 months; 95% CI 12.9–48.3; with, *n* = 67: median OS 51.3 months; 95 %CI 15.2–87.4; *p* = 0.017). The same tendencies could be described in stage I (without, *n* = 5: median OS 60.3 months, 95% CI 0–124.1; with, *n* = 14: median OS 156.1 months, 95% CI 63.1–249.1; *p* = 0.109), stage II (without, *n* = 18: median OS 38.6 months; 95% CI 20.3–56.9; with, *n* = 20: median OS 65.0 months; 95% CI 4.9–125.1; *p* = 0.548) and stage III cohorts (without, *n* = 29: median OS 18.6 months; 95% CI 10.1–27.1; with, *n* = 32: median OS 24.9 months; 95% CI 14.7–35.1; *p* = 0.230). These results are visualized in the Kaplan–Maier curve in Figure 2.

### 3.6. Multivariate Analysis

All parameters that were associated statistically significantly with the OS in the univariate analysis were included in the multivariate analysis (age, histology, tumor location, stage, thyroid replacement therapy, T3 levels according to normal limits). Tumor location (*p* = 0.015, HR = 4.228), age at first diagnosis (*p* = 0.002, HR = 1.231), and thyroid replacement treatment (*p* = 0.015, HR = 0.073) were still statistically significant in the multivariate analysis, while stage (*p* = 0.557, HR = 1.746), histology (*p* = 0.245, HR = 0.243), and T3 levels (*p* = 0.122, HR = 2.368) were not.

## 4. Discussion

Gastroesophageal cancer is a widespread disease and accounts for a large portion of the global cancer mortality [1]. Survival in these patients is poor and little is known about reliable prognostic markers. Since thyroid hormones have been found to convey proliferative and pro-angiogenic effects, it is surmised that they might also play a role in tumor development and progression [3,9]. Thus, the aim of this study was to evaluate the association of thyroid hormones with the outcome in resectable gastroesophageal cancer patients.

### 4.1. Demographic and Tumor Characteristics

Cancers of the upper GI tract are known to be among the leading causes of global cancer mortality. While the diseases still affect more males than females [10], in recent decades a shift from squamous cell carcinomas to adenocarcinomas has been observed [11]. The conjecture behind this shift is that changes in lifestyle in the Western world might affect the tumor incidence, e.g., increased prevalence of obesity, increased prevalence of gastroesophageal reflux disease, decline in the prevalence of *H. pylori* colonization, and improved food processing [12,13,14]. The results of these combined trends are also reflected in the data obtained in this study. Hence, we observed: (i) an almost equal number of gastroesophageal junctions (32.5%) and gastric tumors (39.9%), which is opposite to the Asian populations, where the gastric cases are in the majority; (ii) 70.2% of the included patients in this study were male and (iii) the majority (78.7%) of the observed carcinomas were adenocarcinomas. Furthermore, the observed median age of the included patients of 64 years falls within the limits of the already known epidemiology seen in the Western world [13]. Thus, our study population is a large European cohort that is representable and fits current data.

### 4.2. Thyroid Hormone Status

For over a century, the association of thyroid hormones and malignant diseases has been suggested, yet it is still a widely discussed topic of grave scientific and clinical interest [15]. Although several in vitro and in vivo studies investigated the influence of thyroid parameters on upper GI cancers, clinical data on the subject is still scarce. So far, we have already discussed the association of elevated fT4 levels with poorer overall survival of patients with gastroesophageal cancer in advanced stages [7]. However, to our best knowledge, there are no studies discussing this issue in resectable settings. Thus, the obtained data of this analysis sheds new light on this relationship and is an important impulse for further investigations.

Although no statistically significant difference could be identified comparing the thyroid status in the patients’ history with the OS, there was a clear tendency toward the association of hypothyroidism with a favorable and hyperthyroidism with a poor outcome (Figure 2).

Most available literature agrees with this result [3,16,17] and this finding supports the hypothesis that thyroid hormones might have some tumor-promoting properties, possibly by some non-canonical pathway [5,18]. Several recently published articles by various research groups have demonstrated that this signaling pathway is initiated through integrin αvβ3, which is expressed on cancer cells as well as proliferating endothelium [19,20]. Thus, it is surmised that not only the effect of elevated thyroid hormones on cancer cells themselves but that also the dysregulation of angiogenesis plays a major role in thyroid hormone-stimulated cancer progression. Particularly for gastroesophageal cancer, inhibition of angiogenesis in resectable as well as palliative settings is intensively studied and demonstrates promising outcomes [21,22]. However, it is suggested that not all patients obtain benefit from these treatment strategies and hence, an appropriate selection based on biomarkers would be helpful. In regard to these findings, it is surmised that targeting thyroid hormone actions could present a new biomarker and therapeutic strategy against cancer proliferation and angiogenesis [23,24]. As new treatment options are desperately needed in cancer entities with poor survival rates and limited therapeutic strategies, such as gastroesophageal carcinomas, this approach is of high clinical interest. However, no clinical trials are yet underway to explore these potential targets, as the in vivo effects and regulation of these pathways are still scarcely researched.

Furthermore, the prevalence of known hyperthyroidism in our study cohort (5.7%) is exceedingly high compared to epidemiological prevalence in the literature [25]. This finding underlines the possible role that thyroid hormones play in carcinogenesis [26] and has also been investigated by other study groups [27].

In addition, thyroid hormone substitution was statistically significantly associated with the OS in our study cohort (Figure 2). It is of note that most research on the influence of thyroid hormone replacement therapy on tumor progression has been conducted in patients who developed hypothyroidism under cancer-specific therapy, and to the best of the authors’ knowledge none has yet been conducted in gastroesophageal cancer. Since thyroid replacement therapy is one of the most commonly prescribed drugs in Western countries [28], this finding is of high clinical interest. The rationale behind these results might be the different components of thyroid supplements and physiological thyroid hormones [29].

Levothyroxine, which is the most commonly used thyroid supplement, is a synthetically generated substitute of T4 and thereby does not include T3. Thus, the serum levels of T4 might be higher in patients with replacement therapy than in normal individuals, and the supplementation may not result in appropriate serum T3 concentrations [30]. The compound of the supplement might be of high clinical interest, as T3 is discussed to be a more potent promoter of carcinogenesis due to its role as the key regulator of essential cellular processes including proliferation, differentiation, apoptosis, and metabolism [31]. In addition, it is surmised that deiodinases, which activate thyroid hormones by converting T4 into T3 (type I and II) or inactivate them by degrading (type III), might play an important role in relation to malignant diseases. This tight intracellular control of thyroid hormone availability might be dysregulated in patients with cancer and might promote tumor development and progression [32,33]. A recent study by Lin Hung-Yung et al. showed that reverse T3 (rT3), which results from the conversion of T4 to T3, may be a host factor supporting cancer growth [34]. However, the potential involvement of rT3 within gastroesophageal tumorigenesis as well as the exact mechanisms of the interaction between deiodinases and cancer cells are still unknown. Yet it is surmised that the replacement therapy with levothyroxine might disrupt some carcinogenic pathways stimulated by deiodinases.

Thus, levothyroxine is widely discussed to be associated with a reduced risk of cancer. The association with a statistically significantly reduced relative risk of colorectal cancer was recently described [35]. However, this phenomenon has never been addressed in gastroesophageal cancer and other studies even suggest an association with increased cancer risk [36]. Therefore, the results of this study are of grave scientific interest and give an important impulse to further analyze the association of thyroid replacement therapy with upper gastrointestinal cancer.

Furthermore, when we analyzed whether the thyroid hormone values were within, above, or below the limit of normal a statistically significant difference in median OS could be observed for different T3 levels in the overall cohort. This correlation has also been described by several other research groups in different cancer entities [37,38,39] and might also be associated with the nonthyroidal illness syndrome (NTIS), which was recently found to be connected with malignant diseases [40].

However, only few patients had elevated or decreased levels and therefore these results might not be representable. This might be due to the time point of our measurements being the time of first cancer diagnosis when most patients with a history of thyroid disease were already treated and therefore the levels were mostly within the normal limits.

Further investigation of the role of these parameters as prognostic markers needs to be conducted in the form of prospective studies in order to minimize missing values and potential arising biases.

### 4.3. Strengths and Limitations

The strengths and limitations of the analysis need to be addressed. The homogenous study population represents a large European cohort with resectable upper gastrointestinal cancer. The best possible treatment for each patient was ensured by the individual decision of an interdisciplinary tumor board according to the respective standard of knowledge at the time of diagnosis and all patients were followed up regularly.

As this study was conducted retrospectively, the main limitation is missing data values. Due to obligatory and standardized laboratory monitoring, as well as history taking at the Vienna General Hospital, most results were retrievable from the medical records. Still missing endocrinological parameters are listed in Appendix A. Thus, to minimize missing data values and confirm the results of our analysis a prospective study should be conducted. Furthermore, a more detailed examination of laboratory values in a prospective setting might be able to identify the underlying cause of the observed associations. Moreover, longitudinal measurements of endocrinological parameters could give more detail about the influence of thyroid hormones throughout disease progression and should, therefore, be conducted in further studies.

## 5. Conclusions

In conclusion, the results of this retrospective single-center analysis indicate that a history of hypothyroidism, as well as thyroid hormone replacement therapy with levothyroxine, might be associated with a beneficial outcome in patients with resectable gastroesophageal cancer. These results are in line with our results in patients with advanced stages and other studies in various other cancer entities. The role of thyroid hormones in carcinogenesis is still not fully understood and a topic of grave scientific interest and potential clinical relevance. Thus, our results are an important impulse for further prospective studies concerning the association of thyroid hormones with the outcome in patients with tumors of the upper gastrointestinal tract.

## Figures and Tables

**Figure 1 cancers-13-05050-f001:**
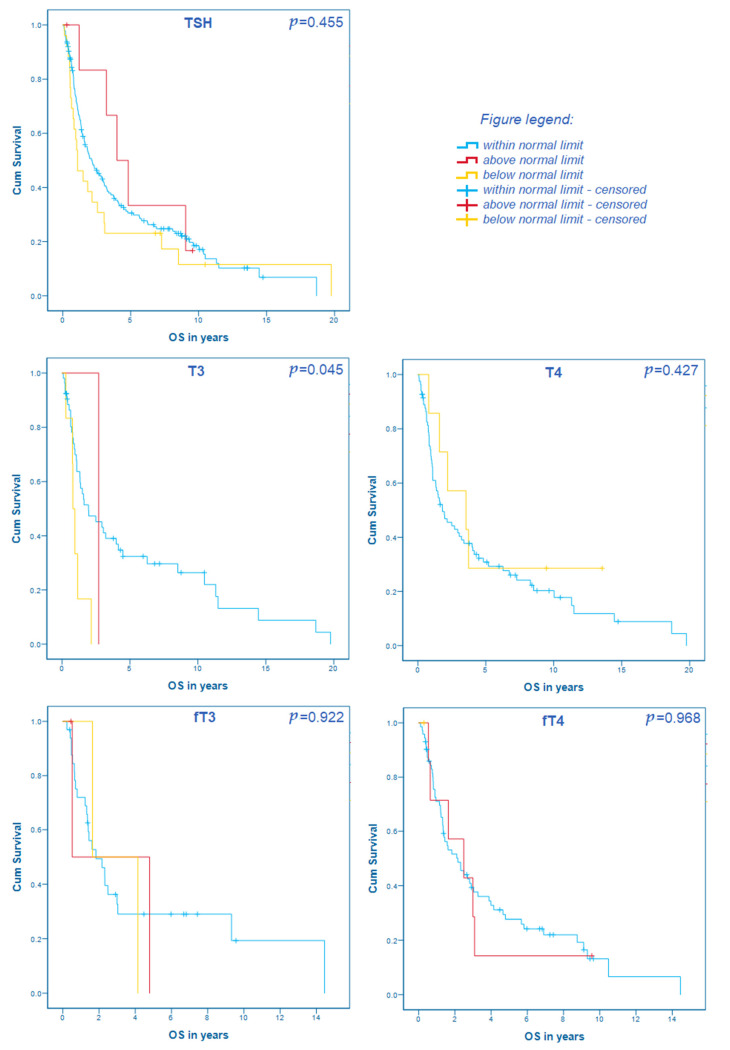
The correlation of thyroid hormone levels within, above, and below normal limits with the overall survival in patients with resectable gastroesophageal cancer.

**Figure 2 cancers-13-05050-f002:**
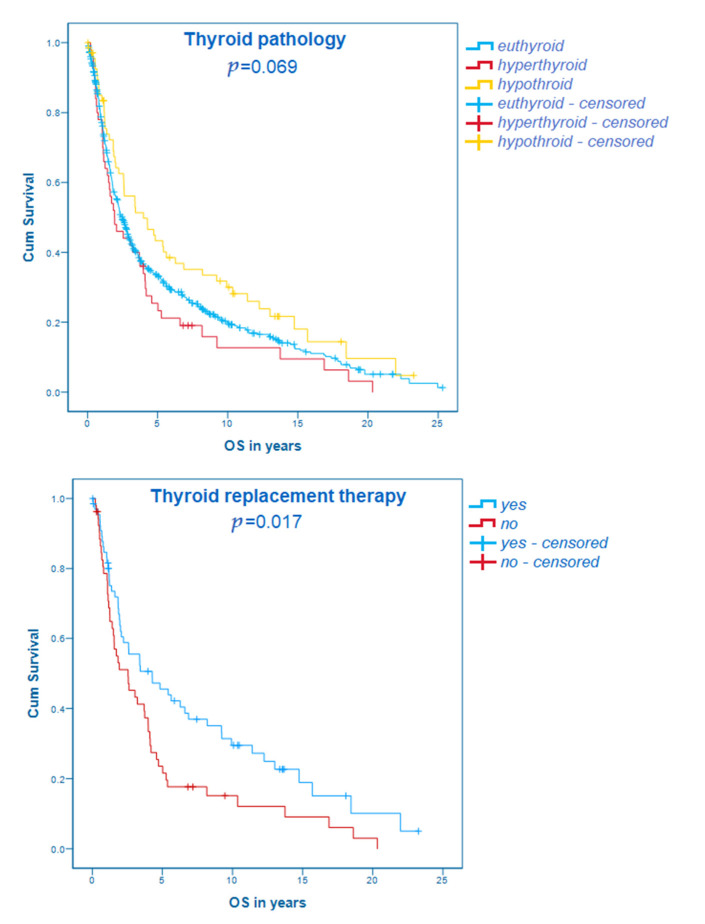
Kaplan–Meier curves concerning the thyroid hormone status as well as thyroid replacement therapy.

**Table 1 cancers-13-05050-t001:** Patient demographics and cancer characteristics and their correlation with survival. Abbreviations: OS—overall survival, CI—confidential interval, HR—hazard ratio.

Characteristic	Value	*p*	OS in Months (95% CI)/HR
**Gender [*n* (%)]**		0.766	
male	607 (70.2%)		31.4 (27.0–35.8)
female	258 (29.8%)		27.9 (23.0–32.8)
**Age (years)**		**<0.001**	1.013 (1.006–1.020)
median (min, max)	64 (28; 92)		
**Body mass index (BMI)**		0.247	0.985 (0.961–1.010)
median (min, max)	24.8 (13.6; 50.0)		
**Localization of cancer [*n* (%)]**		**0.005**	
Gastric	345 (39.9%)		32.4 (25.5–39.3)
Gastroesophageal junction	281 (32.5%)		26.7 (18.6–34.8)
Esophageal	239 (27.6%)		27.3 (20.9–33.7)
**Histological type [*n* (%)]**		**0.001**	
Adenocarcinoma	681 (78.7%)		31.7 (27.1–36.3)
Squamous cell carcinoma	184 (21.3%)		24.2 (18.2–30.2)
**Stage [*n* (%)]**		**<0.001**	
Stage I	155 (17.9%)		87.6 (65.4–109.8)
Stage II	276 (31.9%)		38.6 (30.2–47.0)
Stage III	422 (48.8%)		20.1 (18.0–22.2)
Resectable, but unknown stage	12 (1.4%)		

**Table 2 cancers-13-05050-t002:** Laboratory thyroid parameters and their correlation with the overall survival. Abbreviation: SD—standard deviation.

	TSH.				T3				T4				fT3				fT4			
Cohort	*n*	Median *n* (SD)	*p*	HR	*n*	Median *n* (SD)	*p*	HR	*n*	Median *n* (SD)	*p*	HR	*n*	Median *n* (SD)	*p*	HR	*n*	Median *n* (SD)	*p*	HR
**Overall cohort**	209	1.24 (1.22)	0.642	0.970	60	1.08 (0.27)	0.252	0.493	89	82.0 (16.93)	0.647	1.003	38	3.04 (0.76)	0.917	0.975	80	1.31 (0.31)	0.631	0.887
**Stage I**	35	1.14 (0.78)	0.299	1.367	6	1.00 (0.16)	0.321	n.m.	11	78.0 (16.54)	0.569	0.983	5	2.9 (1.45)	0.766	0.269	15	1.22 (0.24)	0.860	0.742
**Stage II**	55	1.32 (1.42)	0.588	0.924	19	1.02 (0.36)	0.203	0.360	30	82.0 (17.84)	0.970	1.000	10	3.07 (0.26)	0.564	n.m.	16	1.23 (0.27)	0.157	n.m.
**Stage III**	117	1.30 (1.24)	0.396	0.936	35	1.10 (0.24)	0.243	0.288	47	83.0 (16.03)	0.915	0.999	23	3.03 (0.66)	0.956	0.977	49	1.38 (0.53)	0.407	0.781

n.m. = not measurable. The normal ranges of parameters are: TSH 0.44–3.77 μU/mL, T3 0.8–1.8 ng/mL, T4 58–124 ng/mL, fT4 0.76–1.66 ng/dL, and fT3 2.15–4.12 pg/mL.

## Data Availability

The data that support the findings of this study are available from the corresponding author, A.I.-M., upon reasonable request. Data from this analysis were presented as an E-poster at the ESMO (European Society for Medical Oncology) 2020 congress.

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
