# Peer review of "Thyroid Hormone Replacement Therapy Is Associated with Longer Overall Survival in Patients with Resectable Gastroesophageal Cancer: A Retrospective Single-Center Analysis"

_cancers, 2021, doi:10.3390/cancers13205050_

Round 1

Reviewer 1 Report

The main question addressed by the research is- Does thyroid dysfunction in individuals on / requiring exogenous thyroxine supplementation impact on prognosis of solid cancers?

The topic has been controversial in spite of evidence clinical epidemiological ,preclinical and mechanistic molecular studies demonstrating that T4 interaction with tumor cell and endothelial cells via a thyroid hormone receptor on integrin avbeta3 

The tumor sites reviewed and analyzed have not previously been studied. The data is positive and adds to the evidence connecting hypothyroidism with better prognosis in solid cancers.

Conclusions are consistent with the evidence and arguments and they address the main question posed.
References are not appropriate. There should be discussion/mention with reference to thyroid hormone interaction of low to high levels/concentrations with tumor cells and angiogenesis.

There should also be discussion/mention of type 2 and type 3 Deiodinase and interaction with exogenous T4 

 So the interrelationship of thyroid hormone/s i.e. T4 and T3 is developing and more complex than previously thought .Thyroxine  is the major pro-oncogenic hormone  but recent study suggests that reverse T3 [rT3]may be as mitogenic potent as L-T4 [Lin et al   It is also a metabolic product derived from T4.

Reviewer 2 Report

The current study presents comprehensive analysis of patients from the Division of Oncology at the General Hospital of Vienna. The Authors focused on thyroid parameters and patients outcomes seeking for correlations between these to offer novel prognostic markers. Introduction paragraph is short briefly presenting the clinical background and goal of current study. Methods are appropriately designed and matched to a goal.

The manuscript is written with very good English, data are presented in clear manner, understandable to readers. There are only few minor remarks which are listed below:

Line 119 – printing mistake, missing world: “resectable gastroesophagea” cancer

Subparagraph 3.3 – the first sentences are confusing “were within the normal range” since in the next fragments authors mentioned sth different. What did authors meant? This issue requires clarification to prevent misunderstanding.

Legend of Figure 1/2 – it is really hard to read it, the descriptions and legend should be enlarged.

The current manuscript should be considered for publication in Cancers since the presented study can be valuable for readers interested in endocrinological aspects in cancer progression.
